# Metabolic Profiling Analysis Uncovers the Role of Carbon Nanoparticles in Enhancing the Biological Activities of Amaranth in Optimal Salinity Conditions

Ahlem Zrig [1,2,*] , Abdelrahim H. A. Hassan [3], Shereen Magdy Korany [4] , Emad A. Alsherif [5,*], Samy Selim [6] , Ali El-Keblawy [7], Ahmed M. El-Sawah [8] , Mohamed S. Sheteiwy [7,9] , Zainul Abideen [10] and Hamada AbdElgawad [5,11]

1   Laboratory of Engineering Processes and Industrial Systems, Chemical Engineering Department, National School of Engineers of Gabes, University of Gabes, Gabes 6072, Tunisia
2   Faculty of Sciences of Gabes, University of Gabès, Gabes 6072, Tunisia
3   School of Biotechnology, Nile University, Giza 12588, Egypt; azmeysw@gmail.com
4   Department of Biology, College of Science, Princess Nourah bint Abdulrahman University, Riyadh 11671, Saudi Arabia; smkorany@pnu.edu.sa
5   Botany and Microbiology Department, Faculty of Science, Beni-Suef University, Beni-Suef 62521, Egypt; hamada.abdelgawad@uantwerpen.be
6   Department of Clinical Laboratory Sciences, College of Applied Medical Sciences, Jouf University, Sakaka 72388, Saudi Arabia; sabdulsalam@ju.edu.sa
7   Department of Applied Biology, Faculty of Science, University of Sharjah, Sharjah 27272, United Arab Emirates; elkeblawy76@yahoo.com (A.E.-K.); salahco_2010@mans.edu.eg (M.S.S.)
8   Department of Agricultural Microbiology, Faculty of Agriculture, Mansoura University, Mansoura 35516, Egypt; a_elsawah@yahoo.com
9   Department of Agronomy, Faculty of Agriculture, Mansoura University, Mansoura 35516, Egypt
10  Dr. Muhammad Ajmal Khan Institute of Sustainable Halophyte Utilization, University of Karachi, Karachi 75270, Pakistan; abideen_z@yahoo.com
11  Integrated Molecular Plant Physiology Research, Department of Biology, University of Antwerp, 2020 Antwerp, Belgium
*   Correspondence: ahlem18zrig@yahoo.fr (A.Z.); emad_702001@yahoo.com (E.A.A.)

**Abstract:** Enhancing the productivity and bioactivity of high-functional foods holds great significance. Carbon nanoparticles (CNPs) have a recognized capacity for boosting both plant growth and the efficacy of primary and secondary metabolites. Furthermore, while salinity diminishes plant growth, it concurrently amplifies the production of phytomolecules. To ensure the robust and sustainable production of nutritious food, it becomes essential to elevate biomolecule yield without compromising plant growth. Here, we assessed the CNPs priming on plant performance and metabolites of the glycophyte amaranth (*Amaranthus hypochondriacus*) sprouts at the threshold salinity (25 mM NaCl; i.e., salinity that does not reduce growth but enhances the metabolites of that plant). We measured growth parameters, pigment levels, and primary (carbohydrates, amino acids, organic acids, fatty acids) and secondary metabolites (phenolics, flavonoids, tocopherols). CNP priming significantly improved biomass accumulation (fresh and dry weight) and primary and secondary metabolites of amaranth sprouts. Increased photosynthetic pigments can explain these increases in photosynthesis. Enhanced photosynthesis induced carbohydrate production, providing a C source for producing bioactive primary and secondary metabolites. The priming effect of CNPs further enhanced the accumulation of essential amino acids, organic acids, unsaturated fatty acids, tocopherols, and phenolics at threshold salinity. The increase in bioactive metabolites under threshold salinity can explain the CNP priming impact on boosting the antioxidant activities (FRAP, DPPH, anti-lipid peroxidation, superoxide-anion-scavenger, hydroxyl-radical-scavenger, Fe-chelating and chain-breaking activity in aqueous and lipid phases) and antimicrobial activities against Gram-positive and Gram-negative bacteria and fungi. Overall, this study suggested that threshold salinity and CNP priming could be useful for enhancing amaranth sprouts' growth and nutritional quality.

**Keywords:** amaranth sprouts; bioactive metabolites; carbon nanomaterials priming; threshold salinity

## 1. Introduction

Nanotechnology applications in agriculture can improve crop production and ensure food quality and safety in climate-change-resistant environments. They also have the potential to reduce the environmental impact of agricultural processes, helping to protect the environment and reduce climate change risk. One promising technique is seed nano-priming. It has been demonstrated that nanoparticles (NPs) increase seed development by increasing starch metabolism, regulating hormonal balance, and stimulating Fe uptake in wheat and other plants [1]. Recent studies have suggested that CNPs can be transferred from plant roots via the xylem and vascular bundles to the aerial section of the plant during transpiration [2]. They can also enter cells through pores or channels in the cell wall and membrane through the apoplastic pathway and endocytosis, acting as transporters of materials or molecules inside cells [3]. Nevertheless, various studies have shown that CNPs can help plants cope with unfavorable environmental conditions such as salinity [4]. CNPs can induce plant growth under salinity growth conditions, accumulate bioactive antioxidant compounds, and trigger other key metabolic activities in plants [5]. For example, in broccoli, multi-walled carbon nanoparticle priming increased the rate of photosynthesis and water uptake under salinity growth conditions [6].

Salinity affects crop growth, decreasing agricultural production and crop quality [7,8]. This effect of salinity varies with species, genotypes, and growth stages [9]. Consistently, genotypic diversity may provide useful traits for improving salt tolerance [10]. In contrast to extreme abiotic stresses, low/mild stress conditions can be purposefully induced to increase the bioactive compounds in edible food plants [9]. For instance, it induced various biochemical processes in plants, including nitrogen metabolism, ion homeostasis [11], proline metabolism, and osmolytes accumulation [12]. These increases in bioactive compounds in plants are positively correlated with improving the nutritional and functional quality of plant foods, such as sprouts during plant cultivation, as a promising strategy known as biofortification [13,14].

Amaranth (*Amaranthus hypochondriacus*), a salt-sensitive (glycophyte) herb, possesses significant nutritional potential as a pseudocereal, primarily attributed to its abundant supply of high-quality protein [15]. It serves as an alternative to grains that contain gluten in the diets of individuals with celiac disease. Amaranth seeds are recognized as an affordable source of dietary fiber, protein, and antioxidant compounds, enhancing levels during the sprouting process [16]. Amaranth grain is a promising high-quality food source due to its health-promoting effects, attributed to the presence of bioactive compounds in addition to its nutritional value [17]. In recent years, amaranth seeds' composition and nutritional properties have garnered significant attention. Notably, they contain noteworthy amounts of vitamin C, carotene, iron, calcium, folic acid, and proteins [17].

During the sprouting process, the dietary fiber, protein, and antioxidant compound content of amaranth seeds, which are already recognized as a cost-effective source of these nutrients, undergoes an increase [18]. Studies have extensively examined the production, chemical composition, and nutritive value of sprouted amaranth grain. For example, Colmenares de Ruiz and Bressani [19] observed a decrease in lipids and phytic acid levels, along with increased digestibility and vitamins in sprouts compared to seeds. Moreover, during amaranth seed germination, several studies have demonstrated increased phenolic compounds, anthocyanins, and flavonoids. Extracts obtained from germinated seeds have exhibited enhanced antioxidant activity [20], which was attributed to increased polyphenolic compounds [21] or other compounds such as peptides [22]. Apart from antioxidants, little is known about the other physiological activities in germinated amaranth seeds. Some of these activities, such as antihypertensive properties, are developed by

peptide-free or released peptides through hydrolysis. Amaranth protein hydrolysates and peptides have demonstrated antihypertensive properties in vivo and in vitro [23]. Proteins mobilized during germination give rise to peptides that may possess physiological activity. Aphalo et al. [17] showed that amaranth protein-encrypted peptides are primarily released from the albumin and globulin fractions after 48 h of seed imbibition.

Amaranth plants can thrive at moderate salinity (5000 ppm NaCl, 85 mM NaCl), without compromising their productivity, chloroplast pigments content, leaf photosynthesis rate, transpiration rate, stomatal conductance, and activities of antioxidant enzymes SOD, POD, and CAT [24]. However, little is known about the impact of threshold salinity on germination and sprouted amaranth seeds.

Healthy, sustainable food production requires enhancing biomolecule production without reducing plant growth. Plant-based systems could be designed to enable higher biomolecule yields without sacrificing plant growth. One approach for optimizing plant-based systems can be achieved by controlling environmental conditions. Higher salinity enhances biomolecule accumulation but reduces plant growth and production. However, lower salinity can enhance biomolecule production without affecting plant growth and performance. Several studies have examined the individual or combined effects of salinity and CNPs on adult plants' biomass, metabolite accumulation, and activities. However, to date, no studies have explored the combined effects of seed priming with CNPs and threshold salinity on the primary and secondary metabolites in sprouted amaranth seeds. The objective of this study was to evaluate how priming amaranth seeds with CNPs and subjecting them to threshold salinity, both individually and in combination, affect biomass accumulation and the production of bioactive compounds in the plant's antimicrobial activity. We hypothesize that seed priming with CNPs stimulates bioactive compound production in sprouted amaranth under the threshold salinity. The results of the study would enhance the in-door production of healthy, sustainable food from sprouted amaranthus seeds. This would provide a nutritious, low-cost food source for areas facing food insecurity, especially in hot arid regions with salinity and water scarcity.

## 2. Materials and Methods

### 2.1. Carbon Nanoparticles (CNPs) Characterization

CNPs, water-dispersible nanoparticles contain C (63%), O (34%), H (1.6%), and N (1.4%). These NPs (Vulpes Inc., St. Louis, MO, USA) are 20–130 nm (size) and have 35–50 $m^2\,g^{-1}$ (specific surface area) and 7–11% (porosity) with surface negative charges (zeta potential −67.6 mV).

### 2.2. Sprout Production

Seeds of amaranth species were surface sterilized by submerging in a 50% (*v/v*) solution of commercial sodium hypochlorite (2.5 g 100 $g^{-1}$) for 10 min, rinsed once with sterile distilled water. Four groups of amaranth seeds received different treatments (groups): (1) control or distilled water, (2) threshold salinity-treated seeds, (3) CNPs-treated seeds, and (4) CNPs + threshold salinity-treated seeds. A preliminary experiment with different salinity levels defined the threshold salinity as 25 mM NaCl. Above this limit, the biomass of sprouted seeds was decreased. Fifty amaranth seeds were immersed in a 20 mL solution of each treatment. The CNPs were applied in a solution containing 80 μg/mL of CNPs. After 12 h of imbibition, the treated and untreated seeds were germinated on vermiculite trays. According to a pilot experiment with a range of CNPs concentrations (10–80 μg/mL). We selected the most effective concentration that induced seed sprouting and bioactive compounds production.

Germinated seeds were watered (Milli-Q water) twice a week. The Hoagland nutrient solution was supplied in all treatments once at the start of the experiment to nourish the seedling emergence. Trays of germinated seeds were moistened with 10 mL of 25 mM NaCl solution (groups 2 and 3) or deionized water (groups 1 and 3). All treated sprouts were grown in a climate-controlled chamber at 21/18 °C over a 16/8 h day/night photoperiod



(150 µmol PAR m$^{-2}$ s$^{-1}$, with 60% humidity). After ten days of growth, sprouts from each tray were weighed to determine their fresh and dry weight and then stored at 80 °C for further biochemical analysis. Three replicates, each with fifteen sprouts from each tray, were used for each measurement.

### 2.3. Pigments

A MagNALyser (Roche, Vilvoorde, Belgium) was utilized at 7000 rpm for 1 min to homogenize the sprout samples, and then samples were centrifuged for 20 min at 14,000× $g$ and 4 °C. The supernatant was filtered using an Acrodisc GHP filter (0.45 µm 13 mm) (Gelman, Ann Arbor, MI, USA) and further analyzed through HPLC (Shimadzu SIL10-ADvp, Kyoto, Japan, reversed-phase, at 4 °C). The separation of pigments was carried out on a C18 silica column (Waters Spherisorb, 5 µm ODS1, 4.6 × 250 m, at 40 °C), using a mobile phase composed of solvent A (81:9:10 acetonitrile/methanol/water) and solvent B (68:32 methanol/ethyl acetate), at a flow rate of 1.0 mL/min at room temperature [25]. Chlorophyll a and b and β-carotene were detected at 420, 440, and 462 nm using a diode-array detector (Shimadzu SPD-M10Avp, Kyoto, Japan). Concentrations were calculated using Shimadzu Lab Solutions Lite software version 5.71 SP2.

### 2.4. Carbohydrate Analysis

Soluble sugars were separated using ethanol (80% vol/vol) at 80 °C for 60 min. Freshly prepared anthrone reagent (150 mg anthrone in 100 mL H$_2$SO$_4$ [72%]) was then added, and the mixture was heated in a water bath at 100 °C for 10 min. It was then cooled in an ice bath for 5 min. The residual pellet's starch concentration was measured after soluble sugar extraction. The starch solution was hydrated, 90% gelatinized, precipitated, washed with ethanol, centrifuged, and vacuum-dried at 30 °C before being extracted with amylase and amyloglucosidase. Total soluble and insoluble sugar were calculated using a multi-mode microplate reader (Synergy Mx, Biotek, Santa Clara, CA, USA) by measuring their absorbance at 625 nm [26].

### 2.5. Amino Acids Analysis

The analysis of amino acids was conducted using the method described by [27]. In this procedure, 100 mg of each amaranth sample was homogenized with 5 mL of 80% ethanol at 5000 rpm for 1 min. Following centrifugation at 14,000× $g$ for 25 min, the resulting supernatant was resuspended in 5 mL of chloroform. The residue was then extracted using 1 mL of H$_2$O. The pellet and supernatant were resuspended in chloroform and centrifuged at 8000× $g$ for 10 min. To determine the retention time of each amino acid, 15 reference standards (0.05 µM mL$^{-1}$ for each) were used. Amino acid determination was carried out using an internal standard, aminobutyric acid. The extracts were then centrifuged for 10 min at 20,000× $g$, and the resulting aqueous phase was filtered through Millipore micro-filters with a pore size of 0.2 µm. The amino acids were quantified using a BEH amide column (2.1 mm × 50 mm) and a Waters Acquity UPLC TQD device. The elution involved two solvents, A (84% ammonium formate, 6% formic acid, and 10% acetonitrile, *v/v*) and B (acetonitrile and 2% formic acid, *v/v*), resulting in the integration of amino acid peaks. The data were processed with Star Chromatography software (version 5.51).

### 2.6. Organic Acids Analysis

To determine the organic acids, amaranth sprouts were extracted in phosphoric acid for 35 min at 4 °C [28]. HPLC detected organic acids through a UV detection system (210 nm). The content of organic acids was quantified using the corresponding standards.

### 2.7. Determination of Fatty Acids Level

Fatty acid content was detected using the method of Hassanpour et al. [29], which utilized GC/MS (MSD 5975-mass spectrometer) to measure the levels of fatty acids. The

concentration of each molecule was determined by comparing the peak area of each chemical to a calibration curve of the relevant standard.

### 2.8. Determination of Tocopherol Content

According to the procedures outlined by AbdElgawad et al. [30], tocopherols were extracted in n-hexane solvent and measured by HPLC (Shimadzu, Kyoto, Japan) under normal phase conditions (Partisil Pac 5 m column material, length 250 mm, i.d. 4.6 mm). Dimethyl tocol (DMT; 5 ppm) was also applied as an internal standard. The data were analyzed using Shimadzu Class VP 6.14 software integrated into the HPLC system.

### 2.9. Determination of Antioxidant Activity

To measure the total antioxidant capacity, FRAP and DPPH methods were used by extracting sprouts in 80% ethanol [31] phosphate buffer (20 mM, pH 7.4) oxidized by OH radical formed by $Fe^{3+}$ reaction with ASC and $H_2O_2$ (1 mM) at 38 °C [32]. The oxidation products were mixed with TBA in perchloric acid at boiling temperature for 15 min and measured at 532 nm in 3:1 of ethanol: ether (*v:v*). Iron-binding, iron-oxidizing, and chain-breaking properties in aqueous phase assay. Antioxidant activities could also be determined by modulating reciprocal concentrations of iron and ascorbate used for DR oxidation. The reaction was carried out at 38 °C for 1.2 h, and then the formation of TBARS was determined as previously demonstrated. Trolox was used as a water-soluble chain-breaking agent. Quercetin, an iron oxidant, was used as a Fe chelator, and transferrin as an iron chelator.

### 2.10. Antibacterial Activity

The antibacterial activity of Amaranthus seed sprout extract was assessed against various microbial strains through the disc diffusion method. The extent of inhibition was gauged using a Vernier caliper to determine their antibacterial activity.

### 2.11. Statistical Analyses

The R statistics program was used for statistical analysis (Gplot, Agricola). One-way analysis (ANOVA) followed by Tukey's test as a post-hoc test ($p \leq 0.05$) was used. The principal component analysis procedure was carried out using the R statistics package. At least three instances (n = 3) were conducted. Using the Corrplot package, Pearson correlation analysis was performed to assess the relations between the different variables.

## 3. Results

### 3.1. Effect of CNP Priming and Threshold Salinity on Growth Parameters

Amaranth sprouts from CNP priming showed noteworthy increases of 23% in both fresh weight (FW) and dry weight (DW) compared to those from non-priming seeds (Figure 1). However, when amaranth sprouts were exposed to threshold salinity, slight reductions in FW and DW were observed, at 13% and 22%, respectively, compared to the control group. Interestingly, the combined application of CNP priming and threshold salinity resulted in marked increases in FW and DW (13% and 11%, respectively) compared with the control group.

### 3.2. Effect of Seed CNP Priming and Threshold Salinity on Sprouts Pigment Content

The application of CNP priming significantly increased ($p < 0.05$) all amaranth sprout pigments as compared to the control (Figure 2). However, threshold salinity significantly increased α-carotene, β-carotene, and lutein contents. However, the threshold salinity slightly decreased the total pigments, Rubisco enzyme activity, and α-cryptoxanthin compared to the control group ($p < 0.05$). Interestingly, the combined application of threshold salinity and CNPs priming significantly increased ($p \leq 0.05$) the total pigments, α-carotene, β-carotene, lutein, α-cryptoxanthin, and Rubisco enzyme activity (reduction of 23%, 33%, 20%, 37%, 32%, and 37% respectively), compared to the control group.

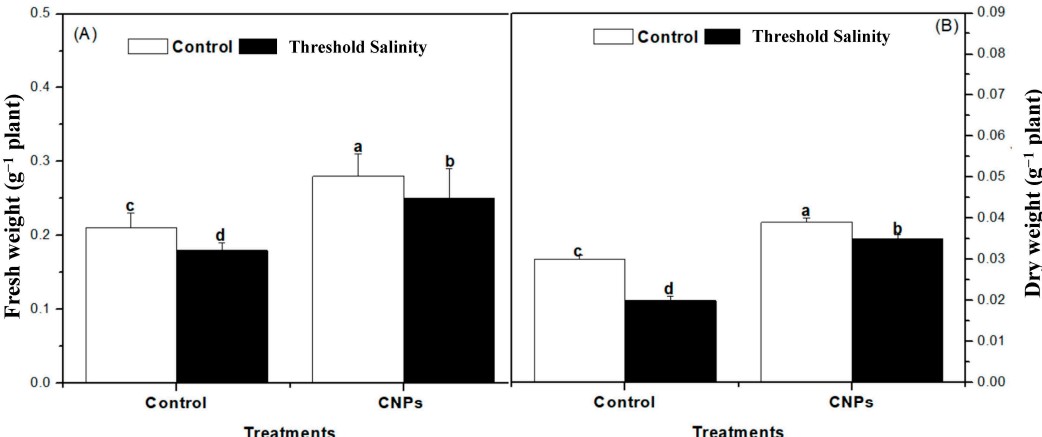

**Figure 1.** Effect of CNP priming and threshold salinity on fresh weight (**A**) and dry weight (g/plant) (**B**) of amaranth sprouts. Data are represented by the means of four replicates ± standard error. Different small letters (a–d) above bars indicate significant differences between control, threshold salinity, and CNP priming ($p < 0.05$).

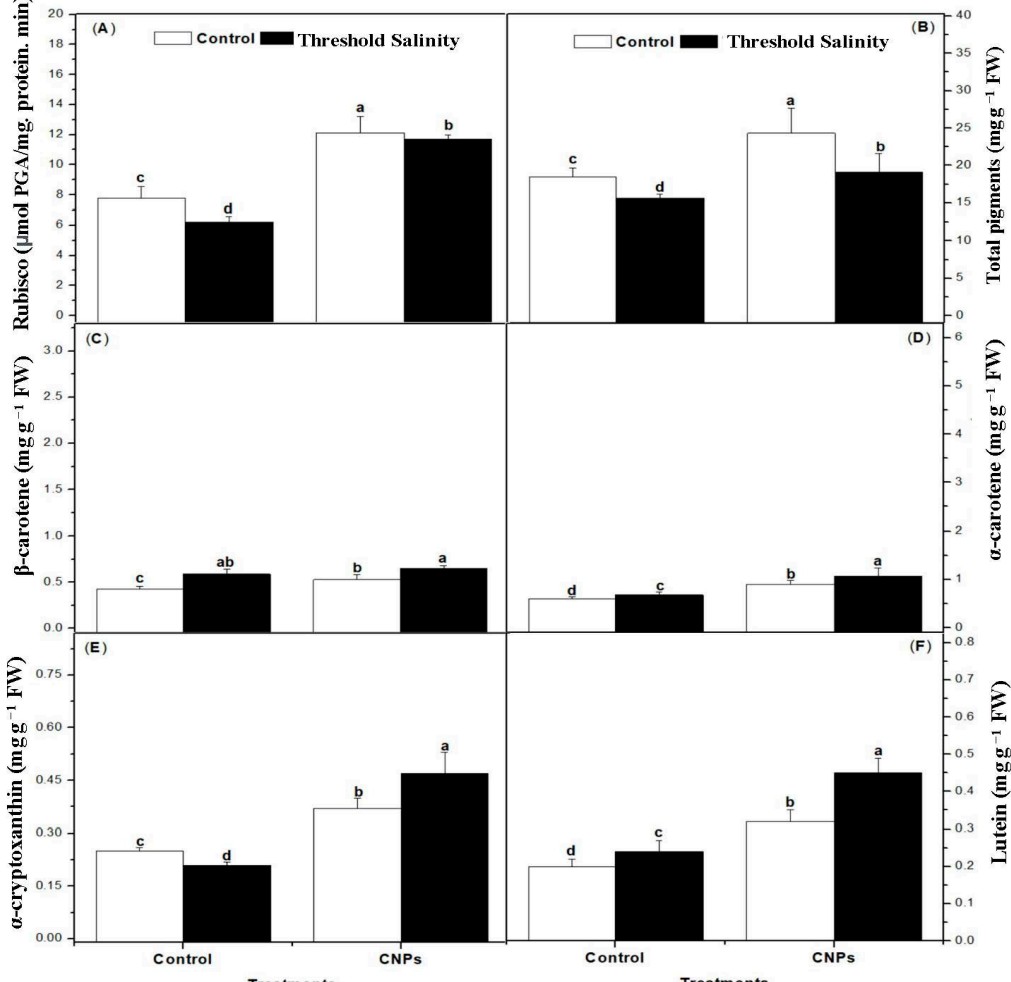

**Figure 2.** Effect of CNP priming and threshold salinity on Rubisco Enzyme activity (umol PGA/mg. protein. min) (**A**), total pigment content (**B**), β-carotene (**C**), α-carotene (**D**), α-cryptoxanthin (**E**), and lutein (mg g$^{-1}$ FW) (**F**) of amaranth sprouts. Data are represented by the means of four replicates ± standard error. Different small letters (a–d) above bars indicate significant differences between control, threshold salinity, and CNP priming ($p < 0.05$).

### 3.3. Effect of CNP Priming and Threshold Salinity on Carbohydrate Production

Various soluble sugars were analyzed in the investigated amaranth sprouts, including sucrose, glucose, and fructose (Figure 3). The application of CNP priming resulted in noteworthy increases in total sugars, sucrose, glucose, and fructose, by 24%, 12%, 25%, and 33%, respectively, compared with the control group ($p < 0.05$). However, threshold salinity significantly reduced the contents of total sugars, sucrose, glucose, and fructose by 22%, 23%, 33%, and 27%, respectively, compared to the control group. Interestingly, the reductions in carbohydrate content resulting from threshold salinity application were significantly improved by CNP priming.

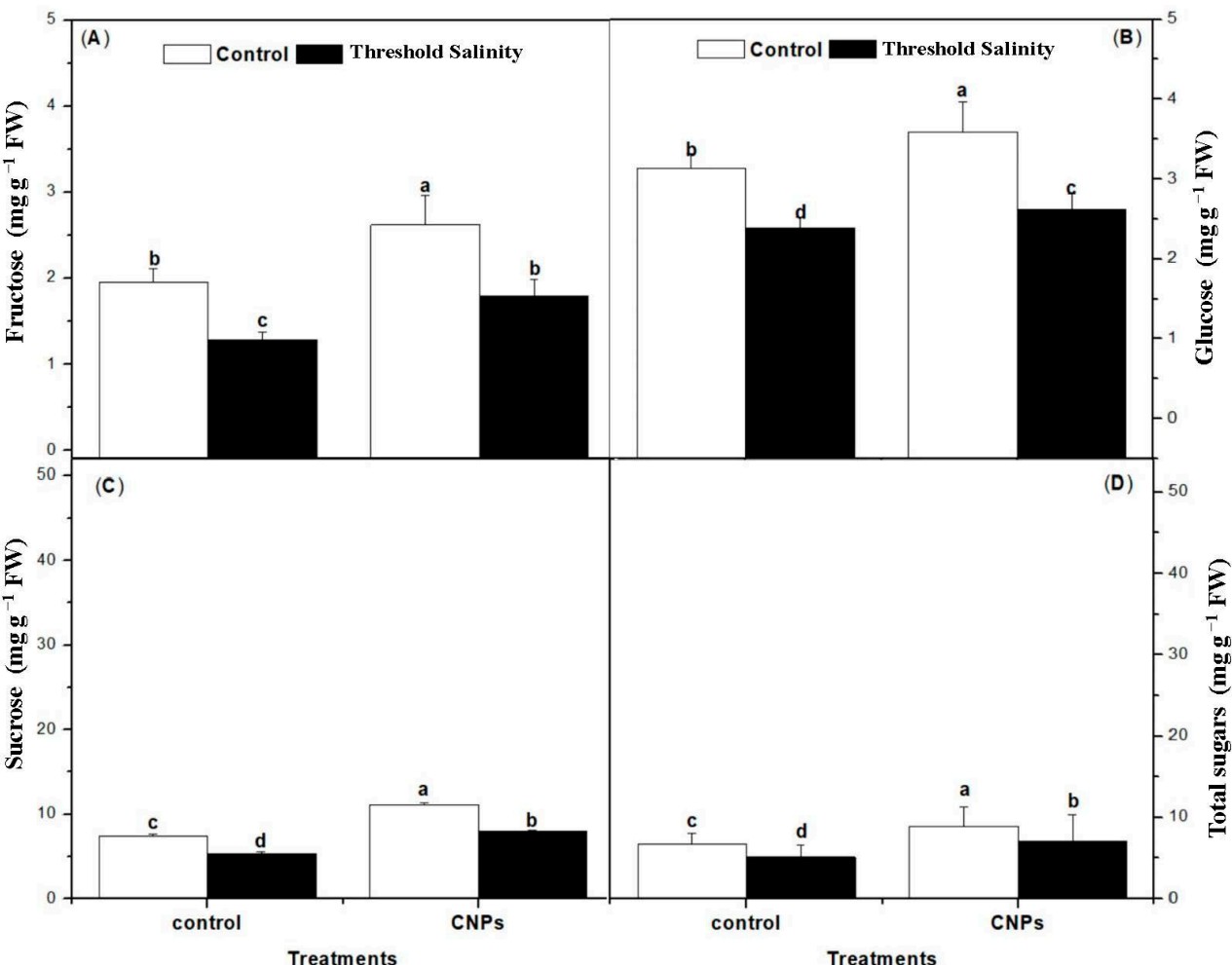

**Figure 3.** Effect of CNP priming and threshold salinity on carbohydrate content (mg g$^{-1}$ FW). Fructose (**A**), glucose (**B**), sucrose (**C**), and total sugar (**D**) in amaranth sprouts. Data are represented by the means of four replicates ± standard error. Different small letters (a–d) above bars indicate significant differences between control, threshold salinity, and CNP priming ($p < 0.05$).

### 3.4. Effect of CNP Priming and Threshold Salinity on Amino Acids Contents

Nineteen amino acids were identified and quantified in amaranth sprouts at varying concentrations (Table 1). Glycine had the highest concentration among all amino acids (91.3 mg g$^{-1}$ FW), followed by alanine (16 mg g$^{-1}$ FW). CNP priming increased the content of 11 amino acids in amaranth sprouts, and the highest increase was recorded in cysteine (61%) and tyrosine (52%), as compared to unprimed sprouts ($p < 0.05$). Threshold salinity either did not affect or slightly increased the levels of several amino acids, including glycine, ornithine, glutamine, asparagine, threonine, valine, serine, phenylalanine, and aspartate, compared to the control ($p < 0.05$). However, the sprouts primed with CNPs exhibited

increases in most of the amino acids, including alanine, glutamine, histidine, arginine, asparagine, methionine, leucine, methionine, glutamic acid, aspartate, cystine, and tyrosine under threshold salinity compared to the effect of CNPs under control conditions ($p < 0.05$).

**Table 1.** Effect of CNP priming and threshold salinity and their interaction on the amino acid content (mg/gFW) of amaranth sprouts.

| Amino Acids | Control | Threshold Salinity | CNPs | CNPs + Threshold Salinity |
|---|---|---|---|---|
| Glycine | 91.30 ± 4.7 b | 112.39 ± 4.04 a | 96.00 ± 4.33 b | 118.8 ± 5.2 a |
| Lysine | 3.85 ± 0.38 c | 5.63 ± 0.61 b | 6.49 ± 1.02 a | 5.83 ± 0.71 b |
| Histidine | 1.73 ± 0.46 c | 2.67 ± 0.63 b | 3.57 ± 0.58 a | 3.89 ± 0.5 a |
| Alanine | 16.04 ± 0.65 c | 18.87 ± 0.7 b | 18.40 ± 0.69 b | 22.1 ± 1.8 a |
| Arginine | 1.43 ± 0.24 b | 1.82 ± 0.7 ab | 2.07 ± 0.20 a | 2.11 ± 0.3 a |
| Ornithine | 0.69 ± 0.13 c | 0.89 ± 0.1 bc | 1.30 ± 0.3 b | 1.93 ± 0.1 a |
| Glutamine | 2.8 ± 0.65 a | 2.54 ± 0.1 b | 2.48 ± 0.01 b | 3.5 ± 0.3 a |
| Asparagine | 1.93 ± 0.20 c | 1.95 ± 0.08 c | 2.87 ± 0.38 b | 4.9 ± 0.3 a |
| Isoleucine | 0.66 ± 0.03 b | 0.60 ± 0.04 b | 1.16 ± 0.04 a | 1.28 ± 0.08 a |
| Leucine | 0.14 ± 0.03 c | 0.21 ± 0.05 b | 0.21 ± 0.05 b | 0.28 ± 0.07 a |
| Methionine | 0.16 ± 0.07 c | 0.22 ± 0.04 c | 0.29 ± 0.15 b | 0.44 ± 0.2 a |
| Threonine | 0.63 ± 0.45 c | 0.50 ± 0.39 c | 1.05 ± 0.67 b | 1.26 ± 0.79 a |
| Valine | 1.26 ± 0.21 b | 1.29 ± 0.08 c | 1.06 ± 0.18 a | 2.1 ± 0.06 a |
| Serine | 1.02 ± 0.40 a | 1.07 ± 0.19 b | 0.97 ± 0.37 b | 1.78 ± 0.5 |
| Phenylalanine | 1.40 ± 0.40 ab | 0.96 ± 0.48 b | 1.50 ± 0.57 a | 2.13 ± 0.2 a |
| Glutamic acid | 1.07 ± 0.05 c | 1.17 ± 0.04 c | 1.44 ± 0.05 b | 1.78 ± 0.2 a |
| Aspartate | 0.22 ± 0.01 c | 0.38 ± 0.0 b | 0.35 ± 0.11 b | 0.48 ± 0.01 a |
| Cystine | 0.27 ± 0.21 d | 0.51 ± 0.40 c | 0.73 ± 0.4 ab | 0.93 ± 0.3 a |
| Tyrosine | 0.46 ± 0.03 d | 0.68 ± 0.06 c | 0.98 ± 0.01 b | 1.6 ± 0.03 a |

Data are represented by the means of four replicates ± standard error. Different small letters (a–d) above bars indicate significant differences between control, threshold salinity, and CNP priming ($p < 0.05$).

### 3.5. Influence of CNPs Priming and Threshold Salinity on Organic Acid Contents

The threshold salinity treatment notably increased the organic acid levels in amaranth sprouts, except for succinic and citric acid, which experienced significant decreases of 66% and 44%, respectively, compared to the control. On the other hand, the application of CNP priming did not significantly affect the contents of most organic acids in amaranth sprouts (Table 2). However, the combined application of CNPs was, and threshold salinity significantly increased malic acid, citric acids, fumaric acids, and succinic acids.

**Table 2.** Effect of CNP priming and threshold salinity on the organic acids content (mg/gFW) of amaranth sprouts.

| Metabolites/Treatments | Control | Threshold Salinity | CNPs | CNPs + Threshold Salinity |
|---|---|---|---|---|
| Oxalic | 3.1 ± 0.17 b | 3.08 ± 0.3 a | 3.1 ± 0.2 ab | 3.67 ± 0.20 a |
| Malic | 14.4 ± 1.02 b | 17.6 ± 1.0 b | 15.2 ± 1.1 b | 29.1 ± 0.4 a |
| Succinic | 6.42 ± 0.27 c | 8.7 ± 0.06 b | 6.4 ± 0.26 c | 11.6 ± 0.6 a |
| Citric | 6.26 ± 0.39 b | 5.49 ± 0.0 b | 2.7 ± 0.15 c | 8.9 ± 0.60 a |
| Isobutyric | 4.02 ± 0.24 b | 5.91 ± 0.3 a | 4.0 ± 0.25 b | 4.9 ± 0.5 ab |
| Fumaric | 0.32 ± 0.02 c | 0.52 ± 0.03 b | 0.3 ± 0.03 c | 0.86 ± 0.4 a |

Data are represented by the means of four replicates ± standard error. Different small letters (a–c) above bars indicate significant differences between control, threshold salinity, and CNP priming ($p < 0.05$).

### 3.6. Influence of CNP Priming and Threshold Salinity on Fatty Acid Contents

In amaranth sprouts, the fatty acid profile analysis revealed that palmitic acid (16:0) and oleic acid (18:1) were abundant fatty acids. Still, palmitic acid was 2-fold lower than oleic acid. The salinity threshold significantly increased saturated fatty acids such as palmitic acid, myristic (C14:0), and stearic (C18:0), but decreased linoleic acid. Still, threshold salinity sharply decreased eicosenoic acid (C20:1), with a 70% reduction compared to the control. Compared with the control, applying CNP priming increased the levels of all saturated fatty acids in amaranth sprouts. Furthermore, the same trends of increase were recorded in four unsaturated fatty acids, and the highest increase was recorded in palmitoleic acid (C16:1), a 50% increase than in the control. Furthermore, the combined application of threshold salinity and CNPs increased the levels of most fatty acids (Table 3).

### 3.7. Effect of CNP Priming and Salinity on Phenols and Flavonoids Contents

Fourteen phenolic acids were present in amaranth sprouts under control conditions. Kaempferol and rosmarinic acid were the most abundant, with 12.8 and 7.5 mg/g, re-

spectively (Table 4). At threshold salinity, a significant increase in all phenolic acids was observed. This increase was 1.7 times higher in kaempferol, luteolin, and naringenin than in the control. Furthermore, CNP priming enhanced the content of all phenolic acids compared to the control condition. However, the combined application of threshold salinity and CNPs did not significantly affect the levels of phenolic acids.

**Table 3.** Effect of CNP priming and threshold salinity on the fatty acids content (mg/gFW) of amaranth sprouts.

| Metabolites/Treatments | Control | Threshold Salinity | CNPs | CNPs + Threshold Salinity |
|---|---|---|---|---|
| Myristic (C14:0) | 0.61 ± 0.01 b | 1.21 ± 0.03 ab | 0.51 ± 0.01 b | 1.43 ± 0.5 a |
| Palmitic (C16:0) | 24.3 ± 2.3 c | 38.1 ± 2.75 b | 14.1 ± 1.43 d | 52.5 ± 5 a |
| Heptadecanoic (C17:0) | 0.06 ± 0.01 b | 0.08 ± 0.0 b | 0.05 ± 0.01 b | 0.15 ± 0.01 a |
| Stearic (C18:0) | 2.3 ± 0.35 b | 2.55 ± 0.1 b | 1.98 ± 0.32 c | 3.94 ± 0.2 a |
| Arachidic (C20:0) | 1.72 ± 0.10 a | 1.12 ± 0.0 b | 0.80 ± 0.04 c | 1.71 ± 0.2 a |
| Docosanoic (C22:0) | 0.78 ± 0.17 a | 0.71 ± 0.03 a | 0.50 ± 0.07 b | 0.92 ± 0.13 a |
| Tricosanoic (C23:0) | 0.04 ± 0.01 b | 0.05 ± 0.0 b | 0.04 ± 0.01 b | 0.13 ± 0.0 a |
| Pentacosanoic (C25:0) | 0.00 ± 0.00 b | 0.012 ± 0.00 a | 0.00 ± 0.0 b | 0.02 ± 0.00 a |
| Palmitoleic (C16:1) | 0.12 ± 0.03 c | 0.16 ± 0.01 b | 0.13 ± 0.04 c | 0.32 ± 0.05 a |
| Heptadecenoic (C17:1) | 0.22 ± 0.06 c | 0.31 ± 0.07 b | 0.09 ± 0.02 d | 0.46 ± 0.06 a |
| Oleic (C18:1) | 52.9 ± 4.6 b | 54.0 ± 2.0 ab | 34.5 ± 13.5 b | 71.1 ± 3.4 a |
| Linoleic (C18:2) | 0.03 ± 0.00 c | 0.05 ± 0.00 b | 0.01 ± 0.00 d | 0.07 ± 0.a |
| Linolenic(C18:3 $\omega - 3$) | 0.02 ± 0.00 b | 0.05 ± 0.00 a | 0.01 ± 0.00 c | 0.06 ± 0.00 a |
| Eicosenoic (C20:1) | 1.37 ± 0.19 c | 1.9 ± 0.12 b | 0.66 ± 0.13 d | 2.5 ± 0.34 a |
| Tetracosenoic (C24:1) | 0.02 ± 0.00 a | 0.03 ± 0.0 a | 0.01 ± 0.00 b | 0.015 ± 0.0 b |

Data are represented by the means of four replicates ± standard error. Different small letters (a–d) above bars indicate significant differences between control, threshold salinity, and CNP priming ($p < 0.05$).

**Table 4.** Effect of CNP priming and threshold salinity on the level of phenols and flavonoids (mg/gFW) of amaranth sprouts.

| Metabolites/Treatments | Control | Threshold Salinity | CNPs | CNPs + Threshold Salinity |
|---|---|---|---|---|
| Gallic acid | 2.32 ± 0.11 b | 3.5 ± 0.18 a | 3.37 ± 0.08 a | 3.17 ± 0.29 a |
| Caffeic acid | 5.0 ± 0.12 b | 7.97 ± 0.34 a | 6.78 ± 0.17 a | 6.51 ± 0.4 ab |
| p-Coumaric acid | 2.31 ± 0.03 b | 3.42 ± 0.17 a | 2.860 ± 0.08 ab | 2.8 ± 0.19 ab |
| Chicoric acid | 2.47 ± 0.02 b | 3.61 ± 0.15 a | 3.080 ± 0.03 a | 3.280 ± 0.04 a |
| Rosmarinic acid | 7.72 ± 0.17 b | 10.4 ± 0.04 a | 8.77 ± 0.19 b | 10.46 ± 0.06 a |
| Protocatechuic acid | 3.25 ± 0.09 b | 4.45 ± 0.08 a | 3.87 ± 0.12 b | 4.36 ± 0.09 a |
| Quercetin | 1.880 ± 0.07 a | 2.6 ± 0.09 a | 2.15 ± 0.09 a | 2.62 ± 0.04 a |
| Naringenin | 0.85 ± 0.02 a | 1.28 ± 0.08 a | 1.09 ± 0.06 a | 1.22 ± 0.04 a |
| Kaempferol | 12.5 ± 0.05 c | 21.6 ± 0.53 a | 15.7 ± 0.93 b | 20.9 ± 0.57 a |
| Luteolin | 2.28 ± 0.09 b | 3.8 ± 0.10 a | 3.10 ± 0.02 a | 3.94 ± 0.12 a |
| Apigenin | 0.07 ± 0.00 a | 0.11 ± 0.00 a | 0.09 ± 0.00 a | 0.11 ± 0.00 a |
| Naringenin | 0.10 ± 0.0 b | 0.16 ± 0.01 a | 0.140 ± 0.01 | 0.150 ± 0.0 a |
| Rutin | 0.03 ± 0.00 a | 0.04 ± 0.00 a | 0.03 ± 0.00 a | 0.03 ± 0.00 a |
| Chlorogenic acid | 0.58 ± 0.02 c | 0.92 ± 0.04 a | 0.75 ± 0.01 b | 0.77 ± 0.06 b |

Data are represented by the means of four replicates ± standard error. Different small letters (a–c) above bars indicate significant differences between control, threshold salinity, and CNP priming ($p < 0.05$).

### 3.8. Effect of CNP Priming and Threshold Salinity on Tocopherol Contents

Our findings indicated a noticeable effect of the salinity threshold on the tocopherol contents in CNP-primed and non-primed amaranth sprouts. The salinity threshold significantly decreased the levels of total tocopherols, α-tocopherol, and β-tocopherol by 21%, compared to the control. Conversely, applying CNP priming increased the total tocopherols compared to the unprimed sprouts. Moreover, the combined application of threshold salinity and CNPs increased the tocopherol contents; the reduction remained minimal compared to the threshold salinity-treated sprouts (Table 5).

**Table 5.** Effect of CNP priming and threshold salinity on the tocopherol (Vit E. mg/gFW) of amaranth sprouts.

| Metablites/Treatments | Control | Threshold Salinity | CNPs | CNPs + Threshold Salinity |
|---|---|---|---|---|
| Alpha-tocopherol | 1.90 ± 0.22 c | 2.7 ± 0.20 b | 2.28 ± 0.10 b | 5.5 ± 0.9 a |
| Beta-tocopherol | 0.32 ± 0.04 c | 0.76 ± 0.05 b | 0.39 ± 0.02 c | 0.91 ± 0.09 a |
| Sigma-tocopherol | 0.08 ± 0.01 c | 0.12 ± 0.0 b | 0.10 ± 0.00 c | 0.18 ± 0.01 a |
| Gama-tocopherol | 0.00 ± 0.00 a | 0.00 ± 0.00 a | 0.00 ± 0.00 a | 0.00 ± 0.00 a |
| Total tocopherols | 2.30 ± 0.27 c | 3.58 ± 0.6 b | 2.76 ± 0.12 c | 6.59 ± 0.9 a |

Data are represented by the means of four replicates ± standard error. Different small letters (a–c) above bars indicate significant differences between control, threshold salinity, and CNP priming ($p < 0.05$).

*3.9. Influence of CNP Priming and Threshold Salinity on Antioxidant Activity*

There was a significant enhancement in the antioxidant activity of amaranth sprouts treated with the threshold salinity. Specifically, salinity increased the FRAP, superoxide anion scavenger and DPPH activity by 38%, 27%, and 37%, respectively, compared to the control group (Figure 4). Similarly, salinity increased the enzymes responsible for inhibiting lipid peroxidation, particularly Fe-oxidizing and chain-breaking activity in the lipid phase, with values exceeding twice those of the control group. A similar trend was observed in all enzyme activities in amaranth sprouts treated with CNP. The combined application of threshold salinity and CNPs significantly amplified both antioxidant activity and enzymes involved in inhibition activity.

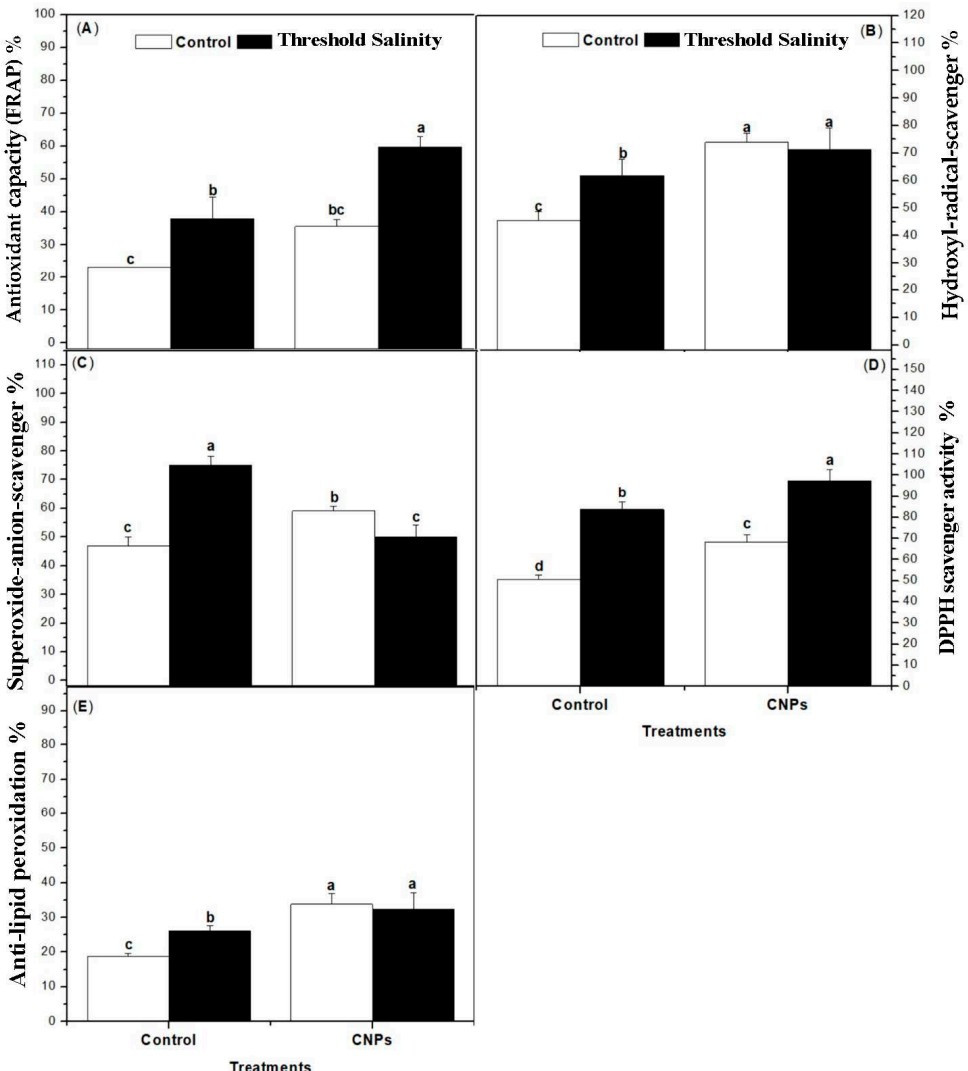

**Figure 4.** Effect of CNP priming and threshold salinity on the antioxidant activity of amaranth sprouts. Antioxidant capacity (FRAP) (**A**), hydroxyl-radical-scavenger (**B**), superoxide-anion-scavenger (**C**), DPPH scavenger activity (**D**), anti-lipid peroxidation (**E**). Data are represented by the means of four replicates ± standard error. Different small letters (a–d) above bars indicate significant differences between control, threshold salinity, and CNP priming ($p < 0.05$).

*3.10. Influence of CNP Priming and Threshold Salinity on Antimicrobial Activity*

The effect of CNP priming and threshold salinity were tested on the antimicrobial activity of amaranth sprout extract (Table 6). Antibacterial activity was tested against a group of Gram-positive and Gram-negative microorganisms. All treatments improved amaranth sprouts' antibacterial activity compared to the control group. However, the

combined application of threshold salinity and CNPs was revealed to have the greatest impact on antibacterial activity.

**Table 6.** Effect of CNP priming and threshold salinity on the anti-microbial activity (zone inhibition diameter (mm)) of amaranth sprouts.

| Microorganism/Treatments | Control | Threshold Salinity | CNPs | CNPs + Threshold Salinity |
|---|---|---|---|---|
| *Staphylococcus saprophyticus* | 13.55 ± 0.75 c | 21.28 ± 1.72 a | 19.64 ± 0.10 b | 21.74 ± 0.84 a |
| *Staphylococcus epidermidis* | 16.96 ± 0.80 c | 24.92 ± 1.60 b | 25.50 ± 0.30 b | 26.85 ± 1.28 a |
| *Enterococcus faecalis* | 21.85 ± 0.88 b | 33.74 ± 0.77 a | 30.77 ± 1.32 a | 22.78 ± 5.53 b |
| *Streptococcus salivarius* | 18.90 ± 0.92 c | 33.34 ± 3.28 a | 21.99 ± 0.97 b | 25.51 ± 1.17 b |
| *Escherichia coli* | 22.90 ± 0.48 c | 27.02 ± 1.59 b | 25.89 ± 0.97 b | 30.70 ± 1.54 a |
| *Salmonella typhimurium* | 26.16 ± 0.86 b | 39.19 ± 0.58 a | 27.30 ± 1.05 b | 34.04 ± 1.20 a |
| *Pseudomonas aeruginosa* | 24.66 ± 1.06 c | 33.54 ± 1.02 b | 28.31 ± 1.22 c | 35.17 ± 1.02 a |
| *Proteus vulgaris* | 22.46 ± 0.85 c | 27.67 ± 3.13 b | 29.45 ± 0.48 a | 29.05 ± 1.42 a |
| *Enterobacter aerogenes* | 14.04 ± 0.63 c | 25.16 ± 0.78 a | 21.50 ± 0.49 b | 21.12 ± 0.88 b |
| *Salmonella typhimurium* | 8.00 ± 0.36 c | 14.22 ± 0.69 a | 11.60 ± 0.26 b | 13.70 ± 0.49 a |
| *Candida albicans* | 9.84 ± 0.41 c | 15.69 ± 0.06 ab | 14.00 ± 0.52 b | 18.49 ± 1.21 a |
| *Aspergillus flavus* | 10.75 ± 0.36 c | 18.56 ± 0.75 a | 14.57 ± 0.75 b | 17.27 ± 0.48 a |

Data are represented by the means of four replicates ± standard error. Different small letters (a–c) above bars indicate significant differences between control, threshold salinity, and CNPs ($p < 0.05$).

### 3.11. Principal Component and Hierarchical Clustering Analyses (HCA)

Pearson's correlation coefficients between biological and chemical factors (Figure 5) indicated that the biomass accumulation of amaranth sprouts strongly correlated with all primary metabolic, such as amino, leaf pigment, and carbohydrate content. However, an inverse relationship was noted between the fresh weight (FW) and both fatty acids and organic acids. The heatmap of the HCA (Figure 6) performed on the medium value of triplicate analysis of all studied parameters showed two main groups: A and B. Group A was homogenous because it was only composed of the plant treated with CNP priming without salinity. However, group B clusters presented salinity treatments.

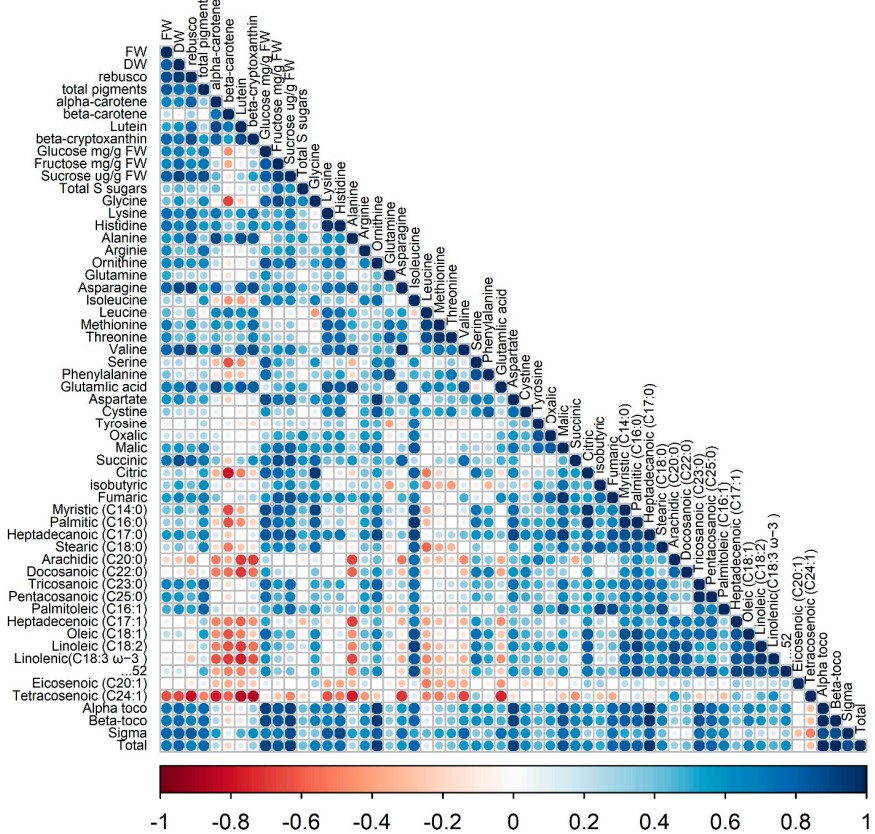

**Figure 5.** The cluster correlation of primary and secondary metabolites of the control and CNPs-primed amaranth sprouts under control or threshold salinity.

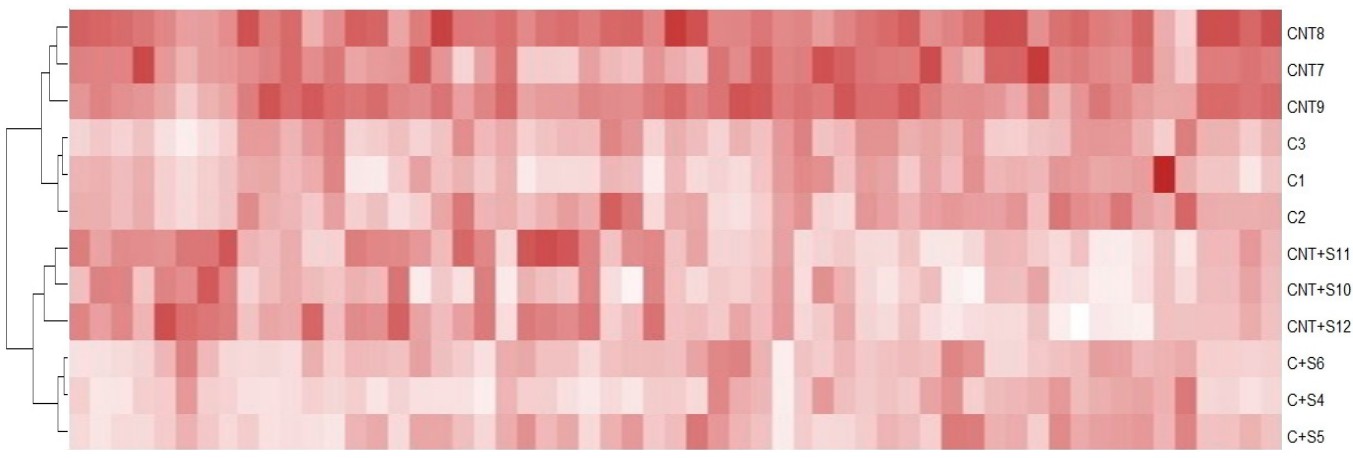

**Figure 6.** The cluster heatmap of growth, photosynthetic parameters, primary and secondary metabolites of control and CNPs-primed amaranth sprouts under control or threshold salinity stress conditions. The graph's horizontal axis shows different treatments for each species and the vertical axis shows different phytocompounds, amino acids, and fatty acids content. Color gradients represent the different values of contents under all treatments compared with that of control. 1 (FW), 2 (DW), 3 (Rebusco), 4 (Total Pigment), 5 (β-carotene), 6 (α-carotene), 7 (α-cryptoxanthin ), 8 (lutein), 9 (Fructose), 10 (Glucose), 11 (Sucrose), 12 (Total sugar), 13 (Glycine), 14 (Lysine), 15 (Histidine), 16 (Alanine), 17 (Arginine), 18 (Ornithine), 19 (Glutamine) 20 (Asparagine), 21 (Isoleucine), 22 (Leucine), 23 (Methionine), 24 (Threonine), 25 (Valine), 26 (Serine), 27 (Phenylalanine), 28 (Glutamic acid), 29 (Aspartate), 30 (Cystine), 31 (Tyrosine), 32 (Oxalic), 33 (Malic acid), 34 (Succinic acid), 35 (Citric acid), 36 (Isobutyric acid), 37 (Fumaric acid), 38 (Myristic acid (C14:0)), 39 (Palmitic acid (C16:0)), 40 (Heptadecanoic acid (C17:0)), 41 (Stearic acid (C18:0)), 42 (Arachidic acid (C20:0)), 43 (Docosanoic acid (C22:0)), 44 (Tricosanoic acid (C23:0)), 45 (Pentacosanoic acid (C25:0)), 46 (Palmitoleic acid (C16:1)), 47 (Heptadecenoic acid (C17:1)), 48 (Oleic acid (C18:1)), 49 (Linoleic acid (C18:2)), 50 (Linolenic acid (C18:3 $\omega - 3$)), 51 (Eicosenoic acid (C20:1)), 52 (Gallic acid), 53 (Caffeic acid), 54 (p-Coumaric acid), 55 (Chicoric acid), 56 (Rosmarinic acid), 57 (Protocatechuic acid), 58 (Quercetin), 59 (Naringenin), 60 (Kaempferol), 61 (Luteolin), 62 (Apigenin), 63 (Naringenin), 64 (Rutin), 65 (Chlorogenic acid), 66 (Alpha-tocopherol), 67 (Beta-tocopherol), 68 (Sigma-tocopherol), 69 (Gama-tocopherol), 70 (Total tocopherols).

## 4. Discussion

### 4.1. CNPs Improved Growth Parameters of Amaranth Sprouts under Threshold Salinity

Plants growing in saline soil face significant challenges. Salinity induces plant metabolic and molecular changes. A previous study showed that salinity can significantly reduce the rate and final percentages of the germination and emergence of amaranth, resulting in uneven stand establishment and decreased crop yields [33]. A growth reduction was observed in the leaf of amaranth sprouts under saline conditions; such a reduction was associated with a reduction in photosynthetic pigments [34]. Agricultural productivity and quality are negatively affected by salt stress, which puts a strain on the food supply to meet the demands of a growing population [35]. Indeed, the results of this study demonstrated that 25 mM NaCl had an insignificant negative impact on the fresh and dry weights of both the shoots and roots of amaranth sprouts, supporting a previous study conducted by Hossain et al. [36] on adult plants. Furthermore, salt stress reduced the growth characteristics of amaranth weights of the shoot and root, and the leaf area [37].

In the present investigation, the CNP priming mitigated the slight negative effect of salinity on the growth parameters of amaranth sprouts. It has been reported that seed priming with CNPs can enhance survival rates, promote growth and development, and increase biomass production while improving plant response to salt stress [38]. CNPs have also been shown to upregulate genes encoding aquaporin proteins, thereby promoting the formation

of water channels (aquaporins) in seed coats [39]. These effects ultimately increase growth rates and productivity and enhance plant salt tolerance [40]. In our study, applying CNPs mitigated threshold salinity, stimulating beneficial plant reactions. Consistent with induced growth, CNPs improved the synthesis of photosynthetic pigments [41] and chloroplast development and protection [42]. According to previous research, CNMs positively affect the production of photosynthetic pigments, including chlorophylls, carotenoids, and lycopene [43]. Here, under threshold salinity, CNP priming induced a greater improvement in sprout pigment content. This is particularly important during seed priming, as chloroplasts are essential not only for autotrophic growth but also for seed germination, and their function is critical under abiotic stress.

It has been reported that CNPs increase the expression of genes that control cell division and cell wall extension. Furthermore, CNPs increased water uptake by aquaporins in seed coats, consequentially improving plant productivity [39,44]. Our findings suggest that priming with CNPs improved adaptive low-salinity responses and enhanced plant performance. The application of CNP priming in seeds seemed to interact with threshold salinity and induce a moderate activation of stress response pathways, which can trigger physiological and biochemical changes that enhance nutrient uptake, photosynthesis, and carbon allocation, resulting in increased biomass production and crop yields.

### 4.2. CNPs Enhanced Primary Metabolites Accumulation in Amaranth Sprouts under Threshold Salinity Condition

In contrast to the threshold salinity effect, CNPs are known to enhance the photosynthesis rate [42]. Consequentially, the increase in photosynthesis induces sugar production. Increased soluble sugars have osmoregulation and antioxidant potential [45]. Sugar production can also contribute to the biosynthesis of antioxidant metabolites by providing the substrate in the oxidative pentose-phosphate pathway. This study observed a similar increasing trend of soluble sugars, glucose, fructose, and sucrose in CNP-primed amaranth sprouts exposed to NaCl stress. Soluble sugars are a crucial organic solute, maintaining cell equilibrium [46]. Furthermore, fructose, one of the sugars in nature, has been acknowledged for its exceptional ability to stabilize the function of proteins by acting as an effective osmoprotectant and osmolyte, thereby providing an environment conducive to protein stability. This is consistent with Hu et al. [47], who found that adding CNPs to *Zea mays* can increase carbohydrate accumulation. This demonstrates their capacity to maintain the osmotic balance and their function in assimilating carbon and nitrogen. Additionally, Sarkar et al. [48] found that salt-stressed *Vigna radiata* accumulated more glucose and fructose when exposed to CNPs.

In addition to their role in plant growth, enhanced carbohydrate levels contribute to other primary (amino acids, organic acids, and fatty acids) and secondary (tocopherols and phenolics) metabolites syntheses by providing biosynthesis energy and precursors. Supporting this hypothesis, threshold salinity and/or CNP treatments induced significant changes in the levels of most detected organic acids and essential and non-essential amino acids and fatty acids. Due to their advantageous nutritional components, including amino acids, the juvenile stages of *Brassicaceae* vegetables, including amaranth, have gained attention as functional vegetables [49]. Meanwhile, applying CNPs through seed priming promotes the accumulation of amino acids in amaranth sprouts with or without salinity conditions [50].

Additionally, amino acids have antioxidant properties in amaranth plants; free amino acids are necessary for secondary metabolism and the production of substances that directly or indirectly impact plant-environment interactions [51]. For instance, asparagine, leucine, methionine, as well as alanine were surprisingly accumulated with CNP priming under threshold salinity, which suggests that photosynthesis may be enhanced under salt stress due to the important roles of amino acids in the control of cytoplasmic pH [40]. Therefore, we suppose that CNPs can serve as elicitors or biostimulants for inducing the biosynthesis of amino acids. Furthermore, these increases improved the nutritional value of amaranth

sprouts under these conditions by increasing the amino acid content. At the organic acids level, the application of threshold salinity and or CNPs significantly improved the organic acid accumulation compared with the control. Malic and citric acids are typically utilized as mobile energy sources in plants, particularly in the absence of carbohydrates [52]. On the other hand, the relatively stable fluctuations in the citric acid of amaranth sprouts under threshold salinity may represent a crucial aspect of their adaptive mechanism to cope with these conditions. Several studies have reported that the organic acids participate in energy production by entering metabolic pathways and being converted into intermediates, such as acetyl-CoA, which is used in the citric acid cycle (also known as the Krebs cycle) to generate ATP, the body's primary energy source [53].

To cope with the salinity condition, plants could modify the fluidity of their membrane lipids by altering the saturation levels of polyunsaturated fatty acids. In this study, the saturated fatty acids palmitic acid and stearic acid, as well as unsaturated fatty, oleic acid (18:1), and eicosenoic, were major compounds [54]. Interestingly, the CNP priming of amaranth seeds under salinity conditions resulted in the accumulation of unsaturated fatty acids, particularly oleic acid, reducing the negative effects of salinity on the plant's growth and development. As in our findings, Yang et al. [55] discovered that the increase in unsaturated fatty acids in the membrane lipids of *Suaeda salsa* provided greater protection to the photosystem II (PSII) under high salinity conditions [56]. The increased unsaturated fatty acids in the membrane lipids could improve the membrane's fluidity, activating the ion channel and protecting the photosystem.

### 4.3. CNPs Improved Secondary Metabolites Accumulation in Amaranth Sprouts under Threshold Salinity

Lipid antioxidants (tocopherols) and water-soluble antioxidants (flavonoids and phenolic acids) are among the most important bioactive phytochemicals [57], being natural antioxidants having a powerful free radical scavenging capacity. Many reports have suggested that applying CNPs in seed priming initiates various physiological and metabolic responses. According to our results, the threshold salinity increased the contents of tocopherols in amaranth sprouts; CNP priming seemed to improve the accumulation of these compounds. Our results suggest that some mechanisms are involved in acclimating amaranth sprouts to saline environments, including the activation of the biosynthesis of tocopherols by CNP priming. In this regard, we suggest that when plants are subjected to abiotic stresses, it is possible for the presence of CNPs to either synergistically or antagonistically interact, causing adverse responses in plants. The improved levels of phenols and flavonoids in CNP-primed sprouts could be attributed to the abundance of C intermediates that could be used for the biosynthesis of these phytochemicals, given that CNPs and salinity have been reported to affect C metabolism. The impact of CNPs on vitamin E (tocopherols) in plants is poorly studied, whereas limited studies have reported improved vitamin E contents in $eCO_2$-treated medicinal plants [58].

### 4.4. CNPs Improved the Nutritional Value of Amaranth Sprouts under Threshold Salinity

The functional food value of plants has been assumed to be associated with their levels of secondary metabolites [59]. Sprouting amaranth plants have been recommended for medicinal uses as antioxidant, antispasmodic, and antimicrobial agents. Such various biological activities could be due to their richness in flavonoids, polyphenolics, essential amino acids, and vitamins [59]. In fact, the remarkable variety of biological activities observed in amaranth sprouts can be attributed to their high levels of flavonoids, polyphenolics, essential amino acids, and vitamins [60]. These compounds are well known for their antioxidant properties, anti-inflammatory effects, and potential contributions to overall health. The antioxidant activity from flavonoids and polyphenolics in amaranth sprouts has been associated with numerous health benefits, including cardiovascular protection, potential anticancer effects, and immune system modulation. Additionally, the abundance of essential amino acids and vitamins further enhances the nutritional value of amaranth

sprouts, making them a beneficial addition to a balanced diet [61]. Herein, CNPs enhanced sprout antioxidant and antibacterial activities, particularly under the salinity threshold.

Combining CNPs and threshold salinity significantly enhanced amaranth sprouts' antioxidant and antibacterial activities. Antioxidants effectively bolster their ability to neutralize and eliminate free radicals [62]. Many studies have reported that primed plants showed higher production of antioxidants in salt-treated plants. In line with our findings, Pérez-Labrada et al. [63] reported that copper (Cu) nanoparticle-based priming enhanced antioxidant activities in tomato plants cultivated under saline conditions. CNPs have also shown the ability to increase antioxidant activity. They can potentially enhance the production or effectiveness of antioxidants, where improved antioxidant activity can have beneficial effects on health and the development of new materials with enhanced properties [44]. Moreover, CNPs have demonstrated their efficacy in enhancing the antibacterial properties of amaranth sprouts, potentially contributing to their ability to combat bacterial pathogens, especially under the salinity threshold. The interaction between CNPs and threshold salinity appears synergistic, resulting in an amplified antioxidant and antibacterial response. This enhanced activity may be attributed to the ability of CNPs to improve nutrient uptake, stimulate plant defense mechanisms, and regulate cellular processes. Further research is warranted to unravel the underlying mechanisms and optimize the application of CNPs for improved sprout quality and health benefits.

## 5. Conclusions

Priming with CNPs improved the slight growth reduction under threshold salinity. Furthermore, combining CNPs and the salinity threshold greatly improved the bioactive compound accumulation and biological activity of amaranth sprouts. This interaction improved photosynthesis-related parameters, consequentially increasing primary metabolism (amino and organic and fatty acids). These increases also provided carbon for secondary metabolite accumulation. Overall, seed priming with CNPs under threshold salinity stress is a cost-effective, environmentally friendly, and beneficial technique that can have favorable effects on agriculture.

**Author Contributions:** Conceptualization, A.Z.; Methodology, H.A., A.Z.; Software, A.M.E.-S.; Validation, A.E.-K.; Formal analysis, E.A.A., M.S.S.; Investigation, A.H.A.H. and S.M.K.; Resources, S.S. and Z.A.; Data curation, H.A., A.H.A.H., S.S., S.M.K., M.S.S. and Z.A.; Writing—original draft, H.A., Z.A; Writing—review & editing, H.A., A.H.A.H., S.S., S.M.K., M.S.S. and Z.A., E.A.A.; Visualization, E.A.A., M.S.S.; Supervision, S.S., H.A.; Funding acquisition, S.M.K. All authors have read and agreed to the published version of the manuscript.

**Funding:** Princess Nourah bint Abdulrahman University Researchers Supporting Project number (PNURSP2023R214), Princess Nourah bint Abdulrahman University, Riyadh, Saudi Arabia.

**Institutional Review Board Statement:** Not applicable.

**Informed Consent Statement:** Not applicable.

**Data Availability Statement:** The data presented in this study are available upon request from the corresponding author.

**Acknowledgments:** The authors would like to thank Princess Nourah bint Abdulrahman University Researchers Supporting Project Number (PNURSP2023R214), Princess Nourah bint Abdulrahman.

**Conflicts of Interest:** The authors declare no conflict of interest.

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
