# Peer review of "Metabolic Profiling Analysis Uncovers the Role of Carbon Nanoparticles in Enhancing the Biological Activities of Amaranth in Optimal Salinity Conditions"

_sustainability, doi:10.3390/su151914650_

Round 1
Author Response
Reviewer 1
Thesis “Metabolic Profiling Analysis Uncovers the Role of Carbon Nanoparticles in Enhancing
the Biological Activities of Amaranth in Optimal Salinity Conditions” analyzed growth
parameters, pigment levels, primary (carbohydrates, amino acids, organic acids, fatty acids) and
secondary metabolites (phenolics, flavonoids, tocopherols) of Amaranthus hypochondriacus. The
role of carbon nanoparticles on the enhanced the accumulation of essential amino acids, organic
acids, unsaturated fatty acids, tocopherols, and phenolics at threshold salinity is also presented.
Due to its novelty, methodology, quality of writing, significance and attitude, this research
can be accepted for publication after minor revisions.
Response: Thanks for positive response
Detailed comments are as follows:
In figures 1-4 is not explain wat means small letter “d”. Explains are only for “a-c”.
Response: Thanks, explanation of letter d is added to figure and tables
In figures 2 what means big letter A-F? Please explain.
Response: Thanks, explanation of letters (A-F)
In figures 4 what means small letters Bc from A and what means big letter A-E? Please
explain.
Response: Thanks, explanation of letters (A-E), The 'bc' letter signifies that it shares a similarity with the values in columns have letters 'a' and 'b.'.
What’s the innovation of this article? The author should explain.
Response: Thanks, innovation of this work is already mentioned, and it is further extend at the end of the introduction
Reviewer 2 Report
The authors studied effect of CNPs on biological activities of amaranth with both physical and chemical quantitative methods. Since the clear meanings of this manuscript to agricultural production and the clear presentation of results, we would recommend its publication after addressing the following comments:
1. In line 236, is it "a-d" instead of "a-c"?
Author Response
Reviewer 2
The authors studied effect of CNPs on biological activities of amaranth with both physical and chemical quantitative methods. Since the clear meanings of this manuscript to agricultural production and the clear presentation of results, we would recommend its publication after addressing the following comments:
In line 236, is it "a-d" instead of "a-c"?
Response: Thanks, added
Reviewer 3 Report
First of all, I have to commend the authors for a very interesting paper. The authors have a lot of results, all of which require a lot of time spent in the laboratory.
This research refers to the seed priming with carbon nanoparticles and threshold salinity on primary and secondary metabolites in sprouted amaranth (Amaranthus hypochondriacus) seeds. The subject is interesting and in agreement with the journal topic. I recommend the revising of the article based on some points mentioned below.
- Please change in all manuscript “hours” with “h” and “minutes” with “min” (lines 101, 151, 152, 163, 165, 166, 175, 178, 181).
- Line 128: please replace “and 35-50” with “and have 35-50”.
- Line 168: please replace “30 degrees Celsius” with “30 oC”.
- Line 208: “ethanol: ether mass ratio”.
- Line 225: “3.1. Influence of CNPs Priming…”.
- Please replace in all figures “thresholdsalinity” with “threshold salinity”.
- Line 270: please replace “B, C, D, E and E” with “B, C, D, E and F”.
- Line 333: please replace ” The was a significant” with ” It was a significant”.
- Please resize the Figure 6 in accordance with the manuscript text.
- Line 508: “balanced diet”.
Some of the references are more than ten years since published. May consider using more current literary sources as much as possible.
Author Response
Reviewer 3
First of all, I have to commend the authors for a very interesting paper. The authors have a lot of results, all of which require a lot of time spent in the laboratory.
This research refers to the seed priming with carbon nanoparticles and threshold salinity on primary and secondary metabolites in sprouted amaranth (Amaranthus hypochondriacus) seeds. The subject is interesting and in agreement with the journal topic. I recommend the revising of the article based on some points mentioned below.
- Please change in all manuscript “hours” with “h” and “minutes” with “min” (lines 101, 151, 152, 163, 165, 166, 175, 178, 181).
Response: Thanks, replaced.
- Line 128: please replace “and 35-50” with “and have 35-50”.
Response: Thanks, added
- Line 168: please replace “30 degrees Celsius” with “30 oC”.
Response: Thanks, replaced.
- Line 208: “ethanol: ether mass ratio”.
Response: Thanks, added
- Line 225: “3.1. Influence of CNPs Priming…”.
Response: Thanks, corrected
- Please replace in all figures “thresholdsalinity” with “threshold salinity”.
Response: Thanks, replaced.
- Line 270: please replace “B, C, D, E and E” with “B, C, D, E and F”.
Response: Thanks, corrected
- Line 333: please replace” The was a significant” with ” It was a significant”.
Response: Thanks, corrected
- Please resize the Figure 6 in accordance with the manuscript text.
Response: Thanks, corrected
- Line 508: “balanced diet”.
Response: Thanks, corrected
Some of the references are more than ten years since published. May consider using more current literary sources as much as possible.
Response: Thanks, recent references are added.
Reviewer 4 Report
The article entitled 'Metabolic Profiling Analysis Uncovers the Role of Carbon Nanoparticles in Enhancing the Biological Activities of Amaranth in Optimal Salinity Conditions' is well presented and posses significant information. A few comments can be incorporated into the revised version.
1. Images of seeds (amaranth species) at different CNP treatments could be included.
2. The reason why the carbon nanoparticles are effective could be included in the discussion section.
3. What is the optimal concentration of carbon nanoparticles used in the studies?
Author Response
Reviewer 4
The article entitled 'Metabolic Profiling Analysis Uncovers the Role of Carbon Nanoparticles in Enhancing the Biological Activities of Amaranth in Optimal Salinity Conditions' is well presented and posses significant information. A few comments can be incorporated into the revised version.
- Images of seeds (amaranth species) at different CNP treatments could be included.
Response: Thanks for suggestion, however we do not think adding images of the seeds will add scientific value to the work
- The reason why the carbon nanoparticles are effective could be included in the discussion section.
Response: Thanks for suggestion, however we do not think adding images of the seeds will add scientific value to the work
- What is the optimal concentration of carbon nanoparticles used in the studies?
Response: Optimal concentration is added and the basis of of this concentration selection is added